

# Flood Volume Allocation Method for Flood Hazard Mapping Using River Model with Levee Scheme

Muhammad Hasnain ASLAM[1,3], Yukiko HIRABAYASHI[2], Dai YAMAZAKI[3], Gang ZHAO[4], Yuki KITA[3], Do Ngoc KHANH[1]

[1]SIT Research Laboratories, Shibaura Institute of Technology, Tokyo, Japan
[2]Department of Civil Engineering, Shibaura Institute of Technology, Tokyo, Japan
[3]Institute of Industrial Science, The University of Tokyo, Tokyo, Japan
[4]School of Environment and Society, Institute of Science Tokyo, Tokyo, Japan

*Corresponding author:* Muhammad Hasnain ASLAM ( hasnain@shibaura-it.ac.jp, hasnain@iis.u-tokyo.ac.jp )

## Abstract

A realistic flood risk assessment is important for rivers where the flood protection infrastructures are dictated by varying return periods. For rivers in Japan, design return periods for flood protection infrastructures range up to 200 years. Large-scale flood hazard mapping increasingly relies on global river models, but these models often lack explicit representation of flood protection levees. In this study, we extend the Global River Model (CaMa-Flood) by integrating levee parameters and applying frequency analysis to simulated flood volumes (the cumulative amount of water exceeding channel storage) and downscaling them to high resolution while explicitly accounting for topographic variability and levee protection.

Levees are represented through heights and fractions, with fractions derived from distance to the river centreline and heights refined by simulations. The method applies both to current simulations, using modelled flood volumes directly, and to future hazard assessment, where frequency analysis of annual maxima provides return-period volumes. These volumes are redistributed to high-resolution unit catchments using terrain data and physically constrained by storage availability.

The results show that integrating levee protection reduces simulated flood volumes, with 10–15% reductions across most return periods in grids containing levees. This reduction reflects the confinement of floodwaters within levee-protected channels, which limits floodplain storage and lowers overbank volumes. At the unit catchment scale, flood extents are also reduced depending on levee fraction and topography. Levees effectively confined floodwaters during moderate to high events, while their influence diminished at extremes where overtopping or volume overestimation became prominent. Findings demonstrate that the levee-integrated downscaling approach captures spatial variability in protection effectiveness, offering a more realistic representation of flood hazard across diverse conditions. By combining hydraulic modelling, frequency analysis, and levee integration, this study provides a comprehensive framework for flood depth mapping, supporting improved resilience planning and basin management.



## 1 Introduction

Flooding remains one of the most recurrent and damaging natural hazards worldwide, with severe socio-economic consequences, particularly in densely populated and low-lying regions. Global flood losses exceeded 1 trillion USD between 35 1980 and 2013 (Dottori et al., 2016), and nearly 290 million people were affected between 2000 and 2018 (Tellman et al., 2021). Flood impacts are expected to intensify under ongoing socio-economic change, climate change, and rapid urbanization (Dottori et al., 2016; Tellman et al., 2021). Hence, flood risk assessment is essential for disaster preparedness and climate adaptation.

Conventional flood hazard mapping is typically based on design precipitation events, where rainfall is first transformed into 40 runoff, then routed to generate design discharges, and finally applied to hydraulic inundation models. Large-scale hydrological and hydraulic models are increasingly being adopted for regional to global flood hazard mapping due to their computational power, advances in remote sensing (Dottori et al., 2016), and the use of global hydrological datasets. A key focus has been integrating hydrological processes with floodplain dynamics to improve accuracy. For instance, Yamazaki et al. (2013) introduced a global river routing model that advanced flood hazard mapping beyond earlier approaches (Pappenberger et al., 45 2012). Hirabayashi et al. (2013) and Kimura et al (2022) used this model to project future global flood risks under climate change using multi-model runoff inputs. Building on these developments, Kita and Yamazaki (2022) applied CaMa-Flood at a national scale in Japan, reproducing broad inundation patterns including backwater and bifurcation effects. However, these global river models often overestimate risk in regions with extensive flood protection, because levees are not explicitly represented, which plays a critical role in modifying floodplain dynamics.

The exclusion of levees in global river models is mainly due to limited levee datasets and the difficulty of integrating such structural controls into large-scale hydrodynamic frameworks. This limitation is particularly critical in high-income countries such as Japan, where levee infrastructure strongly shapes inundation dynamics. A large-scale levee dataset has recently been developed (e.g., Khanh et al. 2025; Zhao et al. 2025). Zhao et al. 2025 introduced a levee representation scheme within CaMa-Flood, enabling the inclusion of levee height and protected fractions in large-scale simulations.

Other issues to overcome in the development of hazard mapping include accounting for spatial variability in floodwater allocation in complex or urbanised floodplains during the downscaling process. Large-scale models typically simulate discharge at a coarser resolution to reduce computational cost and then downscale the calculated inundation extent using high-resolution topographic maps. For example, Kimura et al. (2023) enhanced CaMa-Flood-based downscaling by introducing a reverse-slope correction technique and conducting long-term ensemble flood simulations using GCM-based runoff. Their 60 lookup-based approach provided improved spatial consistency for future flood scenarios under climate change. However, this approach can oversimplify flood inundation dynamics by neglecting the spatial variability and cumulative volume of floodwaters in complex or urbanized floodplains where the presence of structural measures, such as flood protection levees,



plays a critical role in shaping inundation patterns and may mislead the future return-period flood hazard assessment. In particular, downscaling becomes technically challenging when levees are present, since flood volumes must be redistributed across terrain while respecting levee-protected (outside or urban side) and unprotected (inside or river side) zones.

To address this gap, we develop a new downscaling method that explicitly incorporates levee protection when translating flood volumes to inundation depths. The study's main goal was to develop and implement a novel levee-integrated flood hazard mapping framework by: (a) Integrating flood protection levees into large-scale numerical simulations; (b) Incorporating levee information to delineate protected zones within each unit catchment (small, topology-preserving drainage polygons delineated from high-resolution river-network data), and (c) Downscaling coarse-resolution flood volume to high-resolution inundation depth maps by employing terrain-informed volumetric redistribution.

We focus on Japan as a test case because it combines extensive levee infrastructure with hydrological and topographic datasets. Japan's flood protection design standards vary widely across basins, making it ideal for evaluating the influence of levees on flood hazard mapping under diverse conditions. Japan experiences numerous flood events annually, causing substantial damage to infrastructure and loss of human life, due to its steep terrain (Chan et al., 2022), dense population in the floodplain (Huang, 2014), and frequent exposure to typhoons and torrential rainfall (Ridwan et al., 2022). Accurate flood hazard mapping is thus critical for disaster preparedness, urban planning, and the development of effective flood mitigation strategies. Fan and Huang (2020) evaluated flood risk management considering multi-criteria GIS-based analysis, focusing on the Chikuma river basin in Japan. Their study highlighted the importance of spatial exposure and social vulnerability in assessing urban flood risk, but did not incorporate hydrodynamic modelling.

## 2 Data and Methodology

This study proposes a physically consistent, high-resolution flood hazard assessment to address the limitations of downscaling flood depths following outputs from the large-scale CaMa-Flood model. The overall analysis consists of: (1) hydrological simulations at 1-arcminute resolution using the Global Reach-Level A Priori Discharge Estimates dataset (GRADES; Yang et al., 2021) runoff forcing; (2) levee schematisation based on LiDAR-derived levee data; (3) frequency analysis on simulated annual maximum flood volumes (details available in supplementary material) (4) derivation of depth–storage curves per unit catchment based on 1-arcsecond terrain data; and (5) **a hierarchical downscaling algorithm that allocates flood storage volumes to fine-scale pixels, respecting levee-protected and unprotected zones**. The approach is validated using flood extent data from the 2019 Typhoon Hagibis for the Chikuma River in Japan.

### 2.1 Validation Data

Daily river discharge and water level data for validating simulated outputs are obtained from MLIT (Ministry of Land, Infrastructure, Transport and Tourism) for the period from 1980 to 2019, covering over 400 stream gauges across the entire



river basins of Japan. Additionally, we also gathered flood inundation extents from past events, i.e., Typhoon Hagibis, from the Geospatial Information Authority of Japan (GSI 2019) retrieved through
"https://www.gsi.go.jp/BOUSAI/R1.taihuu19gou.html#11" and National land numerical information – hazard maps obtained from https://disaportal.gsi.go.jp/hazardmap/maps/

## 2.2 Large-Scale Flood Modelling

This study utilises the Catchment-based Macro-scale Floodplain (CaMa-Flood) model by Yamazaki et al. (2011) using MERIT Hydro (Yamazaki et al.,2019) as a baseline topography to simulate flood dynamics across Japan. To study the impacts of flood
protection levees on river flood inundation dynamics, we adopted the method of Zhao et al. (2025) to schematize the levee in the Hydrodynamic Model.

The model was implemented at a spatial resolution of 1 arcminute (~1.5 km) for longitudes between 123.0W and 148.0E and latitudes from 24.0S to 46.0N, covering the entire country of Japan. The input runoff forcing was derived from the GRADES dataset, which consists of bias-corrected runoff from the Variable Infiltration Capacity (VIC) land surface model calibrated
against historical observations (Yang et al., 2021). Simulations were conducted from 1979 to 2019, encompassing a 41-year historical data period. Model simulation outputs are generated at a temporal resolution of 24 hours. The simulations were conducted for scenarios with and without a flood protection levee to ascertain the levee's impacts on flood depths and inundation extents.

We confirmed the accuracy of the CaMa-Flood simulations using discharge and water level observations from over 400 stream
gauges across Japan. The model demonstrated strong consistency with observed discharge magnitudes (R ≈ 0.94 for mean and 0.80 for peak flow, supplementary Fig. S3) with most stations achieving satisfactory KGE and NSE values (supplementary Fig. S4 to Fig. S6). The model captured water level fluctuations with moderate to high correlation (R between 0.45 and 0.75, supplementary Fig. S9). These results confirm that the model performs reliably at the national scale. Further statistical distributions and spatial evaluations are presented in the attached supplementary document, section 2.

## 115 2.3 Levee schematisation

In this study, we utilised a recently developed global levee dataset by Khanh et al. (2025), derived from high-resolution (10 m) LiDAR Digital Elevation Models (DEMs). The levee detection method utilised by Khanh et al. (2025) employs four terrain-based parameters—relative elevation, slope, aspect difference, and curvature—to identify levees from DEMs, capturing both artificial and natural embankments. The original 10 m levee data were resampled to a 1-arcsecond (~30m) resolution to align
with our simulation framework. Using this resampled data, we estimated the mean levee distance within each unit catchment (corresponding to the 1-arcminute simulation grid). While resampling from 10m to 30m, a 30m pixel is classified as a levee pixel if any 10m pixels within the 30m block are marked as levee. This resampling is acceptable as all levee segments recorded in Khanh et al. (2025) contain at least 10 pixels.





A Python script was adapted to work with 1-arcsecond resolution terrain and river network data to determine each unit catchment's average levee distance. To incorporate levee protection into the flood hazard mapping framework, we adopted the levee parameterisation method proposed by Zhao et al. (2025), which defines two key parameters that are the levee unprotected fraction (levfrc) and the corresponding levee height (levhgt) for each unit catchment containing a physical levee feature. The levee fraction (levfrc) is calculated as the ratio of the area between the levees (based on the estimated mean levee distance from the river centerline; levdis) to the total area of the unit catchment following Eq. (1). Physically, the levee fraction is the numerical representation of part of the unit catchment that lies inside the levees along the river.

$$levfrc = \frac{rivlen * 2 * levdis}{A_{unit}} \tag{1}$$

We employed an iterative approach based on flood depths derived from CaMa-Flood simulations to estimate levee heights. To determine levee heights, we used basin-specific design return periods compiled from official records maintained by national and prefectural river management authorities. Specifically, our group conducted a literature review of river protection standards covering nationally and prefecturally managed rivers (in total 189 rivers). We did not consider levees in small rivers where the river protection standard data are unavailable. River protection levels were categorized by return periods (e.g., 50-year, 100-year, 200-year), and assigned one representative return period to each river basin as the design standard.

Initially, the model was run without levee protection. Frequent analysis (Gumbel with L-moment) was applied to the simulated annual maximum river water depths to obtain return-period-based depth distributions. These results were then used to assign initial levee heights for each unit catchment by subtracting the corresponding channel depth from the target river water depth (e.g., for the 100-year return period). This initial levee height was incorporated into a new levee-protected simulation. In the second iteration, the model outputs with levee protection were again subjected to frequency analysis to extract updated depths. Based on these, the levee heights were recalculated, and simulations were repeated. This iterative cycle was repeated until the change in flood depth reduction between successive iterations fell below a convergence threshold of 1%. The final iteration showed spatial changes mostly below 0.5%, with one localized peak of 2.5%, indicating that further adjustments would yield negligible improvements. Therefore, the levee heights were considered stable and finalised based on this iteration. Figure 1 illustrates the estimated levee fractions and levee heights for a reach of the Chikuma River.





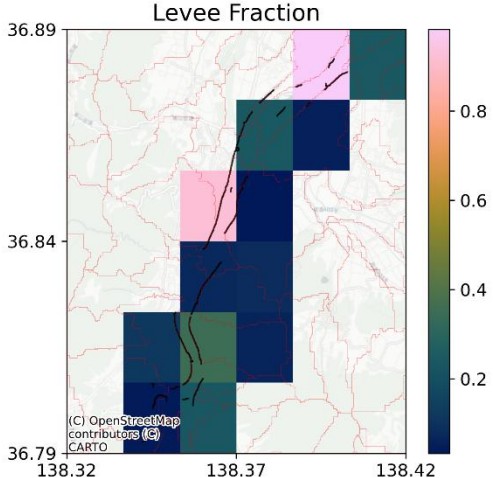 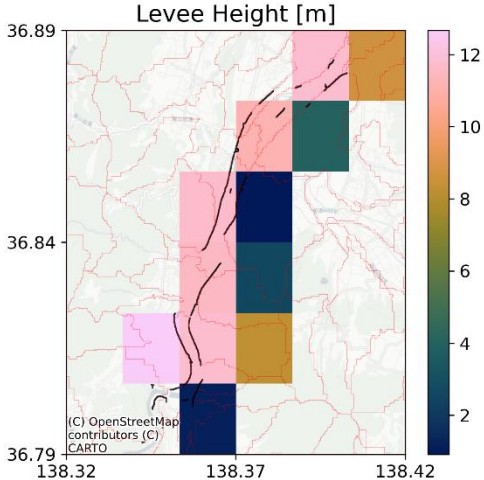

**Figure 1: Estimated Levee fraction (left panel) and Levee height (right panel) for the Chikuma river reach. Black lines represent LiDAR-based levee data, and red polygons are the unit catchment boundaries.**

Note that this uniform assignment does not capture potential spatial variations in protection levels within individual basins. In cases where no levee was present in the levee inventory, those river segments were treated as having no levee protection, even if a design return period had been assigned for the broader basin. This ensures that only physically identified levee structures were considered in the model. It is to be noted that we estimated levee heights iteratively based on simulated flood depths and hence can be underestimated or overestimated than field data.

**2.4    Downscaling Procedure**

Previous downscaling methods in CaMa-Flood spread coarse-resolution flood depths to fine-resolution maps by assuming the water level is uniform across each unit catchment. While this assumption is reasonable in natural floodplains, it does not hold in levee-protected regions where riverside and landside areas can experience different water levels. As a result, the old method can misrepresent inundation depths and extents in such areas. Our new approach overcomes this by including levees in the

downscaling, allowing water levels to vary correctly between levee protected and unprotected areas.

The downscaling method proposed here generally converts coarse-resolution flood volumes into fine-resolution flood depths. This study tested its application by generating high-resolution flood hazard maps from coarse-resolution hydrodynamic model outputs. Specifically, we implemented a volumetric downscaling approach that transforms 1-arcminute storage data into 1-arcsecond inundation depths. The method operates at the unit catchment level, preserving hydrological consistency while

incorporating fine-scale terrain characteristics and levee protection features. For each return period, simulated flood storages with and without levees are determined from CaMa-Flood outputs.



A hierarchical storage allocation strategy was implemented to ensure a realistic flood volume distribution in levee-unprotected and levee-protected areas. The redistribution process follows a priority order that represents natural overtopping behaviour while maintaining volumetric consistency. The storage volume is initially allocated to the low-lying grid cells inside the levee, up to the levee crest elevation. If the volume exceeds the maximum inside-levee capacity, the remaining volume is allocated to grid cells outside the levee, again only up to the levee crest height, utilising the corresponding outside-levee storage curve. If the combined inside and outside levee capacities are insufficient, the volume is distributed across the entire unit catchment, using the total storage curve that accounts for all available terrain.

The resulting water surface elevations are then used to calculate inundation depths by measuring the difference between the surface level and local terrain, ensuring a physical representation of flooding dynamics. This approach offers flexibility and maintains volumetric reliability, making it well-suited for national-scale flood hazard assessments that consider both natural and engineered landscapes.

### 2.4.1    Development of Depth-storage curves for unit catchments

To perform the volumetric downscaling of coarse-resolution flood volumes using high-resolution terrain data, we developed fine-scale, elevation-specific depth-storage relationships that explicitly capture the elevation variability within each unit catchment. These curves provide the foundation for redistributing flood storage across subgrid pixels and estimating inundation depths and extents. The depth–storage curve calculation is universal and can be applied wherever available elevation data. The approach becomes specific to CaMa-Flood when levee-integrated downscaling is performed, since the levee parameters used in the redistribution are consistent with those implemented in CaMa-Flood simulations.

High-resolution topographic data were used to extract the floodplain elevation differences (height of each pixel above the river channel) for all 1-arcsecond grid cells within the unit catchment. These values were sorted in ascending order to represent an idealised terrain profile from low to high ground.

The cumulative flooded area at each elevation step was calculated by summing the areas of grid cells below that elevation, as shown in the schematic below (Fig. 2) and represented mathematically as Eq. (2).

$$A(h_i) = \sum_{j:h_j \leq h_i} Area_j \tag{2}$$



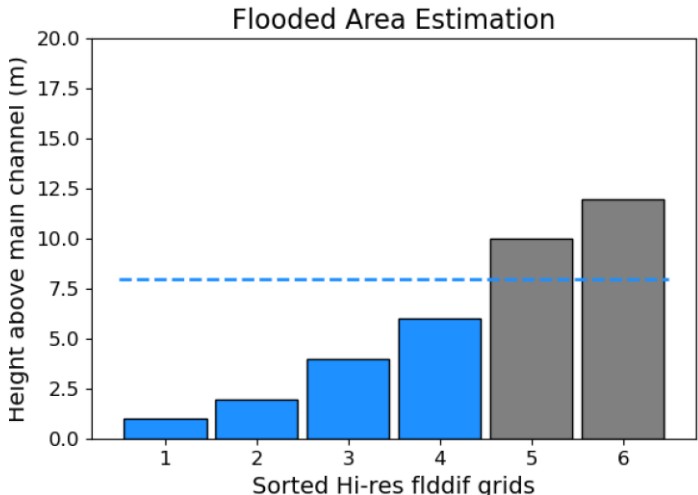

**Figure 2: Schematic illustrating the number of grids considered for inundation area calculation at a specific depth increment.**

Once the cumulative flooded area is computed, the volume of water stored at each depth level is calculated using a trapezoidal method (Eq. (3)). The vertical interval ($\Delta h$; $h_k$ - $h_{k-1}$) corresponds to the difference in successive water surface levels at which cumulative flooded areas are computed. This interval is not constant but varies depending on the fine-scale terrain within each
195 unit catchment, as it is derived from the discrete values representing elevation differences.

$$V(h_i) = \sum_{k=1}^{i} (h_k - h_{k-1}) \cdot A(h_k) \tag{3}$$

This effectively integrates the area under the curve defined by the flooded areas at different water levels, giving the volume at each level (Eq. (4)).

$$V(h) = \int_0^h A(h')dh' \tag{4}$$

V(h) is the volume of water stored at height h and A(h) is the floodplain area at height $h'$. Finally, the derived curve was interpolated to estimate the water surface elevation corresponding to any target volume.

To operationalise the hierarchical storage allocation described before, when levees are present, we partition each unit catchment into levee-unprotected (river-side; inside the levee) and protected (land-side; outside the levee) zones and derive zone-secific depth–storage curves. As illustrated in Fig. 3, separate storage curves were generated for these zones based on the



levee fraction (levfrc) and levee height (levhgt), together with a total curve for the entire unit catchment (excluding channel storage). This ensures flood volume is realistically allocated beyond the unprotected areas when levee capacity is exceeded.

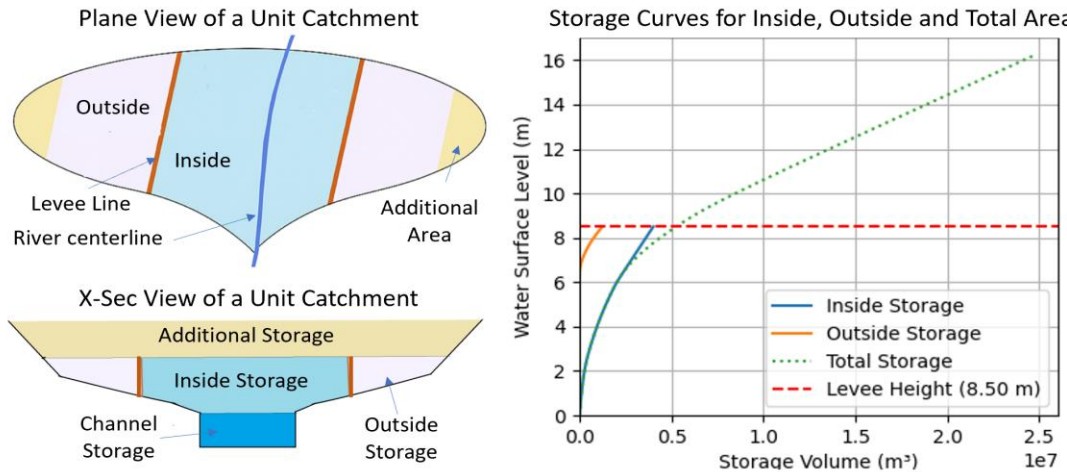

**Figure 3: Estimated depth-storage curves for a unit catchment that contains a levee. The outside storage curve is associated with a pink-shaded zone in the schematic. The inside storage curve corresponds to the light blue zone. The total storage curve is for the entire unit catchment, excluding channel storage.**

It is important to note that when the total storage capacity of a unit catchment is smaller than the simulated or estimated flood volume, the excess volume is accommodated within the same unit catchment by raising the water depth vertically while keeping the inundated area fixed at its maximum extent.

### 2.4.2    Storage-curve–guided redistribution of Flood Volume

The model simulated flood volume in each 1-arcminute grid (representing unit catchment in a High-resolution map) is first adjusted by subtracting channel storage —estimated as the product of river length, width, and depth— to isolate overbank flood volume. All 1-arcsecond pixels within each unit catchment are sorted by elevation and filled incrementally until the total overbank storage is allocated. This filling process is guided by the storage curves derived from high-resolution topography (section 2.4.1), which interpolate the necessary water surface elevation for each catchment to match the target volume. In levee-protected scenarios, the redistribution is refined by dividing each catchment into zones inside (river side) and outside (land side) of levees using the levee fraction (levfrc) and predefined zone classifications (catmz100; pixel-relative locations within unit catchment). Following Eq. (5) and Eq. (6). Separate storage-depth curves are generated for both inside and outside levee areas.

$$inside\ Pixel \Rightarrow catmz100_{ij} \leq levfrc * 100 \qquad (5)$$



$$outside\ Pixels\ \Rightarrow\ \sim inside \tag{6}$$

The downscaling procedure explained above follows a decision-tree approach (Fig. 4), where each unit catchment is evaluated for the presence of levee protection before distributing flood volumes using appropriate storage-depth curves.

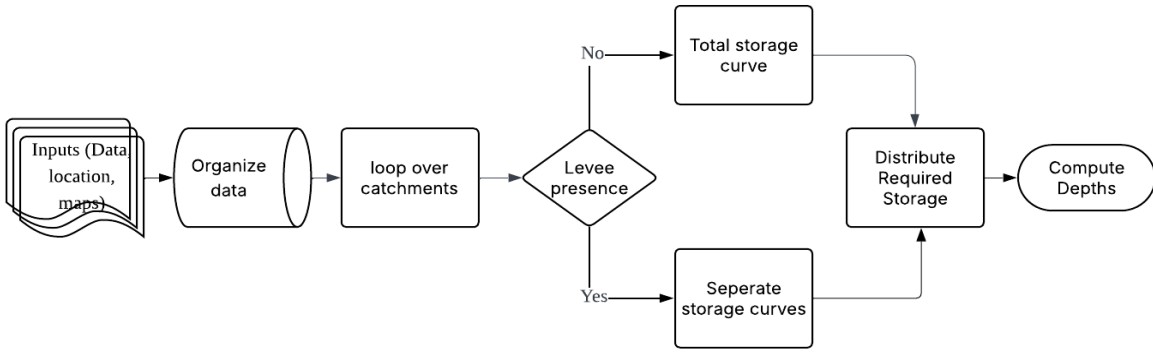

**Figure 4: Downscaling methodology flow chart**

## 3 Results and Discussion

### 3.1 Flood Extents

#### 3.1.1 Comparison with Historic Event

We compared modelled flood extents against observed inundation extent. While comparing flood extents, the river water body was excluded, and the similarity indices (IoU, ACC and FBI) were calculated for floodplains only. The results highlight the benefits of incorporating levee effects into flood modelling, especially in areas where levees dominate flood control (Fig. 5). The model successfully reproduces the general inundation patterns and levee confinement effects observed during the 2019 event in the Chikuma River basin. By redistributing coarse-resolution flood volumes over fine-scale topography while incorporating levee protection, the method closely reproduces the inundation extents and levee-controlled confinement effects observed during the 2019 event. The levee-integration shows notable improvements in replicating observed inundation patterns. The levees effectively contain flooding and capture ponding in topographically vulnerable zones for both upstream [b] and downstream [a] regions.

In the downstream region [a], the simulation with levee information more closely matches the observed inundation, achieving a higher Intersection over Union (IoU = 0.61) and a more balanced Flood Bias Index (FBI = 0.77), compared to the natural scenario. Conversely, in region [b], both scenarios show low agreement with observations, though the levee-integrated case performs better in terms of accuracy. In contrast to region [a], region [b] presents a more complex challenge for accurate flood



modelling. The upper half of this area comprises dense urban settlements, where pluvial flooding due to heavy rainfall and urban drainage capacity can significantly contribute to observed inundation. However, the current modelling framework focuses solely on fluvial flooding from river overflow and does not account for surface water accumulation caused by urban

drainage limitations. This likely explains the low agreement between simulated and observed flood extents in region b, with low IoU values (0.27 for the levee-integrated case and 0.18 for the natural case) and high Flood Bias Indices (especially for natural case).

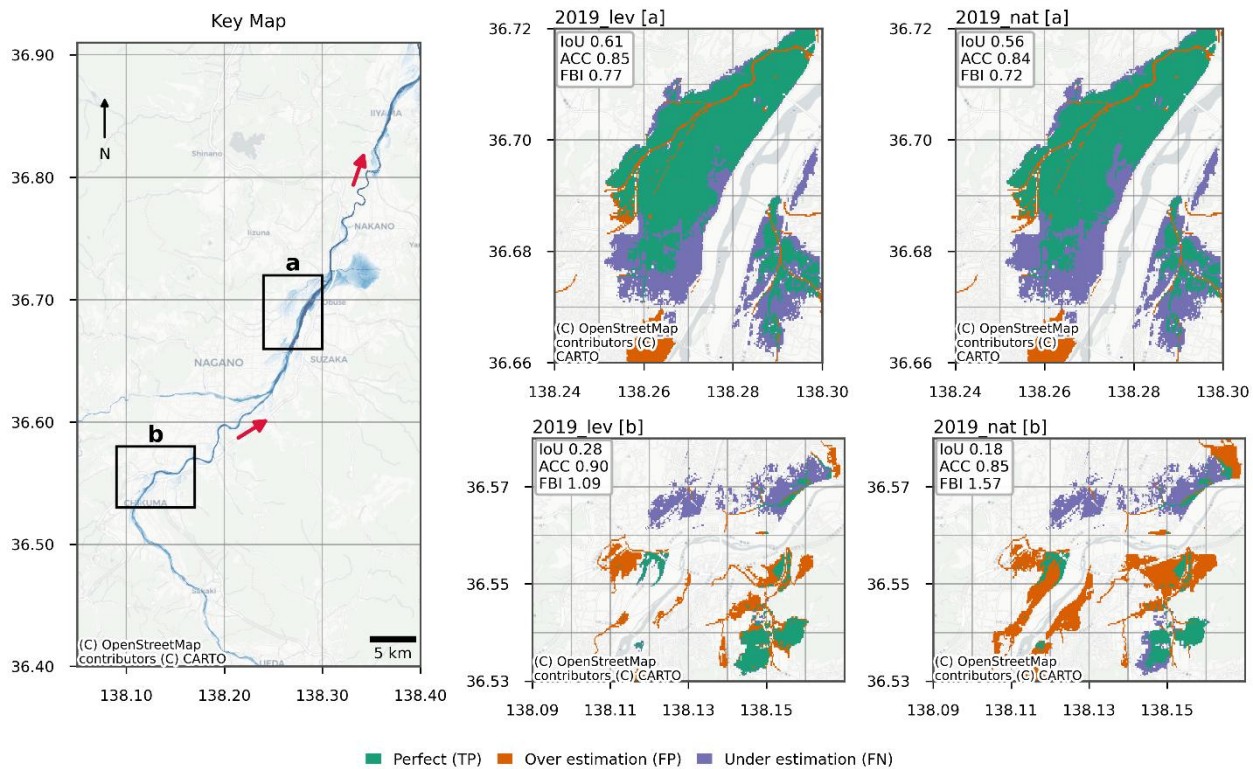

**Figure 5: Estimated Inundation Extents in the Chikuma River. Red arrows indicate the flow direction. Green is the area where**
**simulated and observed inundations are perfectly matched. The modelled flood inundation under neutral and levee-protected conditions in the Chikuma River basin is for two river sections: downstream of the Sai River confluence and upstream of the Sai River confluence. The left panel shows the overall reach with two zoom-in regions labelled [a] and [b] selected for detailed validation.**

Nevertheless, model performance remains sensitive to localized features and conditions. For instance, in the upstream region
[b], the model underestimates inundation in urbanized zones along the left bank, likely due to the absence of pluvial flooding mechanisms and potential underestimation of flood volumes. In contrast, overestimations are observed downstream in low-lying areas along the right bank (see Supplementary Fig. S13), possibly due to topographic limitations in the DEM and the exclusion of small-scale structures such as road embankments that act as barriers to flow. These discrepancies underscore the



importance of high-resolution topography and the inclusion of urban hydrology in future model improvements. While the
levee-integrated simulation slightly improves accuracy, it still overestimates flooding in areas where no substantial river
overflow was recorded.

### 3.1.2    Comparison with City Hazard Maps

Figure 6 compares the 1000-year flood inundation extents estimated by our downscaling approach with the official city hazard
maps of Nagano city (for region [a] in the keymap given in Fig. 5).

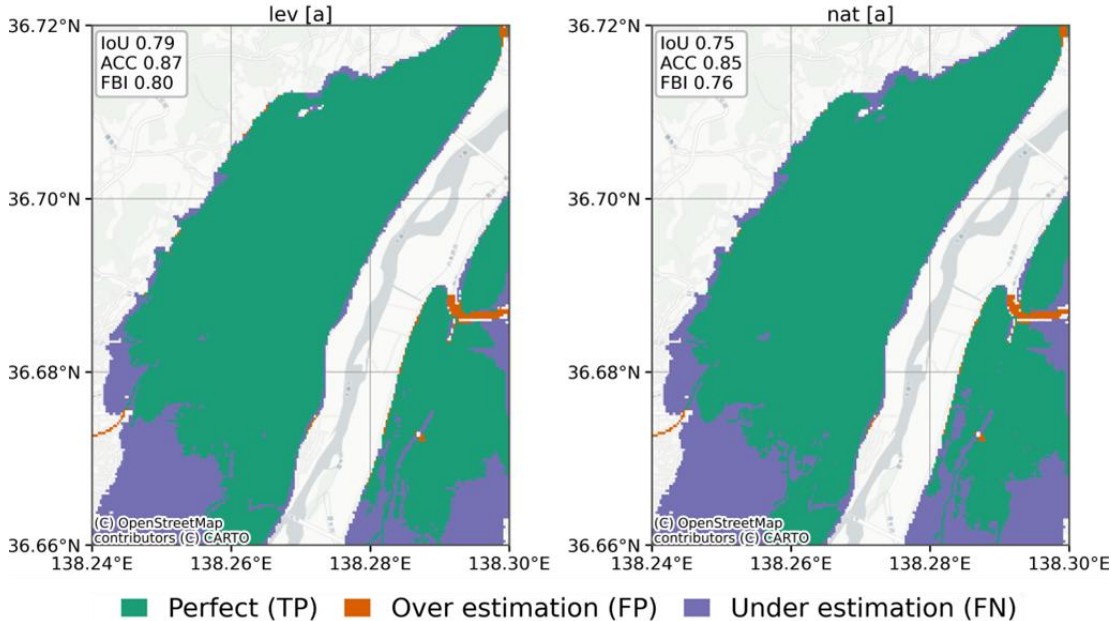

Figure 6: Flood Hazard Maps of the Chikuma River reach for 1000-Year RP compared with the Nagano city hazard map. Green
is the area where simulated and observed inundations are perfectly matched.

City hazard maps were developed using maximum expected rainfall events (once in 1000 years) and included levee
overtopping and breaching assumptions, showing more dispersed flood extents. In contrast, our method is using daily-averaged
runoff inputs, which may fail to capture short-duration peak flows, potentially underestimating flood magnitudes for specific
return periods. Additionally, our model assumes the levee as a static barrier. While this assumption is reasonable for floods
within the design level (around 100-yr), for more extreme events such as the 1000-yr case the levees may no longer be effective,
leading to an underestimation of floodplain inundation. Nevertheless, the proposed framework is flexible and can generate
hazard maps for any return period, enabling comparison with observational or reference datasets whenever such data are
available.



## 3.2    Effect of Levee Integration on Flood Inundation Depths

We further examine the impact of levee integration on flood inundation at a finer spatial scale. Our method calculated the total volume of flood water for each simulation grid (corresponding to a unit catchment of high-resolution maps) and then allocated the flood volume for higher resolution maps. With levee protection, flooding is significantly restricted across the two randomly
selected unit catchments of CaMa-Flood in the case of the 2019 East Japan Typhoon event (Fig. 7). Natural simulation results in widespread inundation across the unit catchments. With levee protection, flooding is significantly restricted. The reduction in flood inundation extent results from topographic patterns and the spatial distribution of integrated levee data.

For the unit catchment A (which is upstream of the confluence with the Sai River and has an estimated levee fraction of 0.39) in Fig. 7, the flood extent without levee protection covered approximately 1.68 km², reduced to 1.04 km² with the inclusion of
levee effects, reflecting a 38.1% reduction in the inundated area. The spatial patterns indicate that levees well protect this area. The unit catchment B, downstream of the confluence (the bottom panels in Fig. 7), with an estimated levee fraction of 0.29, showed a somewhat similar effect of levee integration. Under the natural scenario (bottom-left), the inundated area was 0.98 km², which also significantly reduced to 0.65 km² in the levee scenario (bottom-right), resulting in a 33.7% reduction. The significant reduction in flood extents indicates that the levee system in these catchments effectively confines floodwaters,
underscoring their role in mitigating inundation across diverse settings.



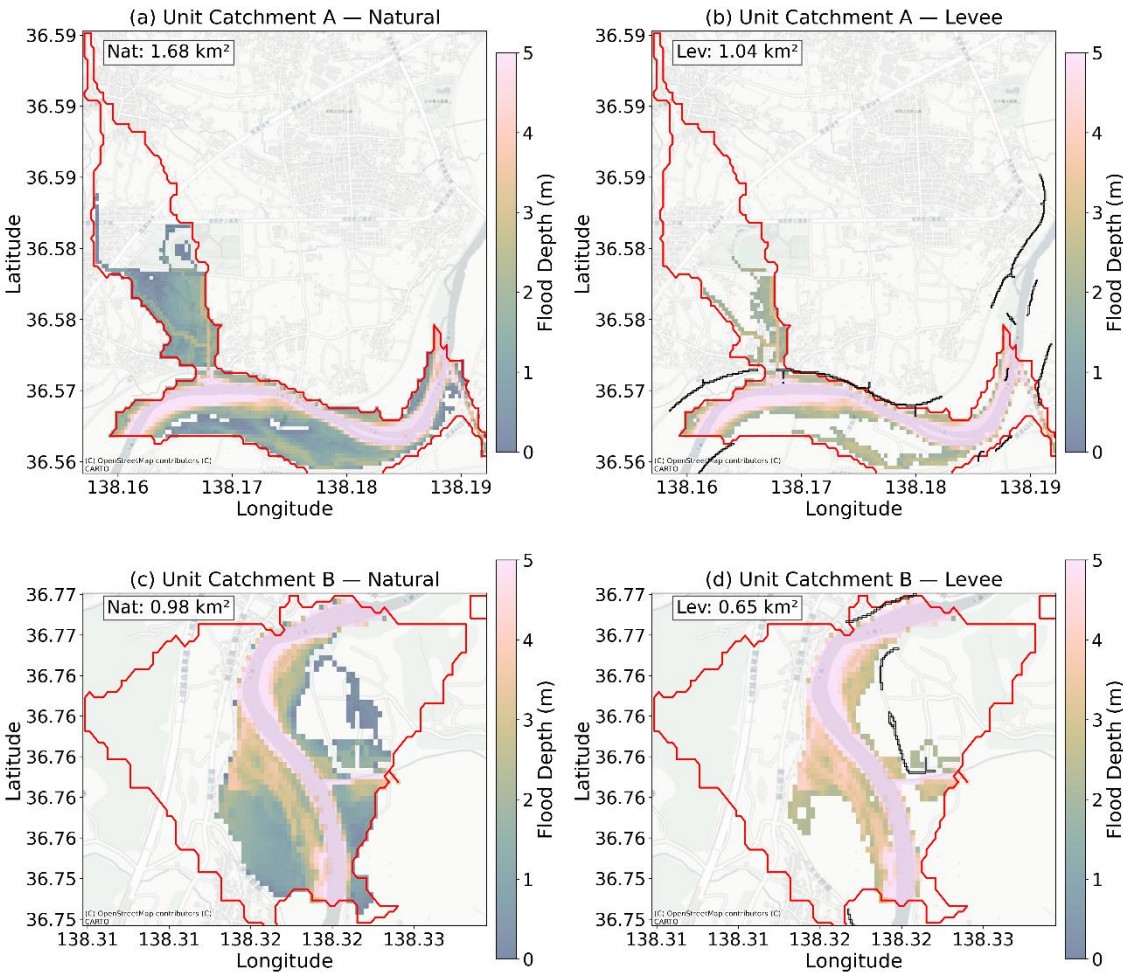

**Figure 7: Spatial distribution of simulated flood depths for Nat (Left Panels) and Lev (Right Panels) scenarios during the 2019 East Japan Typhoon. Red outline is the unit catchment boundary, the black lines represent levee data, and inundation areas are inserted in text boxes at top left**

Similar results are seen in the case of flood extent in different return periods. Figure 8 shows the simulated flood inundation depth maps for a downstream section of the Chikuma River, located approximately 14 km upstream of the confluence with the Sai River. The figure compares natural and levee-protected scenarios across multiple return periods (RPs). At lower RPs (5–50 years), the effect of levees is especially pronounced, with flood areas reduced by up to ~50%. Even under higher return periods (RP 500–1000 years), although less significant, levee protection continues to limit flood spread, demonstrating robust

structural performance under both moderate and extreme flood conditions. Since the design criteria for the Chikuma River is a 100-year return period flood (according to the National database from MLIT, for the reach under consideration), the levee effect for floods with a return period higher than the design return period will be insignificant. Supplementary Fig. S10 and





Fig. S11 illustrate the flood progression in the region 22 km downstream and 14 km downstream of the Sai River confluence, respectively, for natural and with levee cases, with percentage reduction presented in Fig. S12. These findings underscore the spatial variability of levee effectiveness and highlight the importance of site-specific flood risk assessments when evaluating structural interventions.

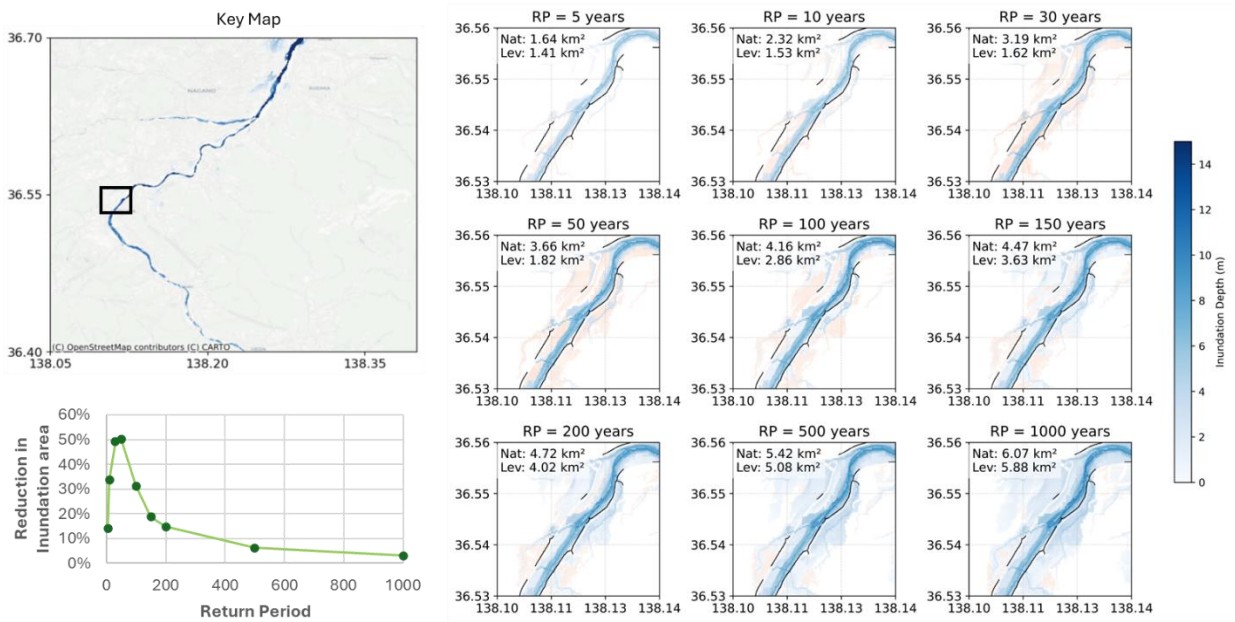

**Figure 8: Simulated Flood inundation depths for the Chikuma River across multiple RPs (~14 km upstream of the Sai–Chikuma confluence) under natural (red) and levee-protected (blue). Black lines represent levee data. A line graph (bottom left) shows the percentage reduction in the inundation area with levee**

## 3.3 Flood Hazard Maps

We developed detailed future flood hazard maps and presented them for the Chikuma River. The flood hazard maps (with levee protection) created for return periods (RPs) of 10 and 100 years (Fig. 9) show a clear progression in inundation extent and severity across the Chikuma River basin. These maps, generated using high-resolution (1-second) downscaling of simulated flood depths, classify flood hazards according to Japanese flood depth thresholds and overlay them on detailed base maps for a realistic spatial context. The 10-year scenario, which reflects relatively frequent flood events, indicates that even common floods pose a significant threat to critical infrastructure and urban areas near the floodplain, especially downstream of the Sai–Chikuma confluence, where terrain and flow convergence increase water accumulation.

As the return period increases to 100 years, both the extent and depth of flooding increase notably. The hazard footprint broadens significantly into urbanised areas. Flood hazard magnitude and spatial extents grow nonlinearly with the return period. Areas at risk of extreme-depth inundation increase dramatically beyond RP 100, while shallow and moderate flood



classes (0.5–5 m) consistently dominate lower RPs. This pattern suggests the prioritisation of preparedness and infrastructure adaptation in these zones.

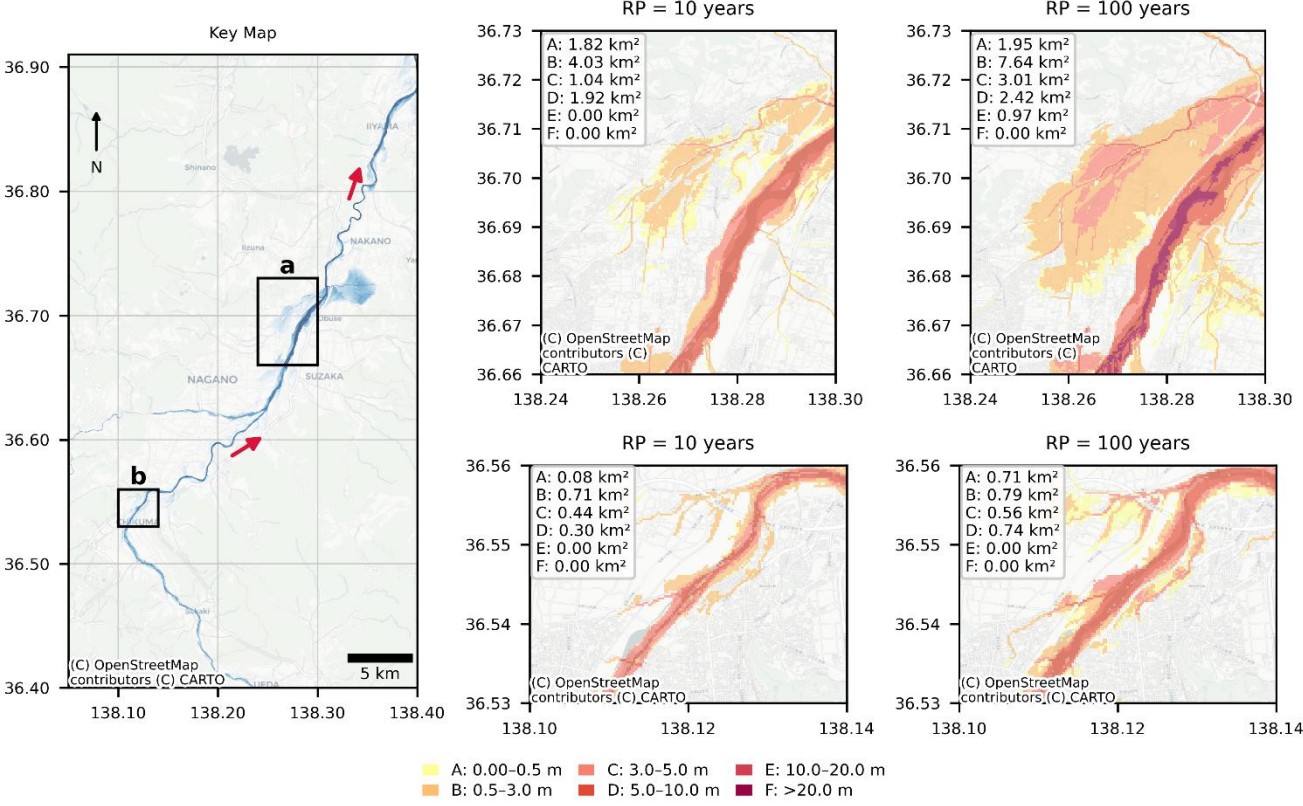

**Figure 9: Flood Hazard Maps for the Chikuma River for 10 and 100-Year RPs with levee scenario (The inundation area of each Hazard class is written in each text box of subplots. The hazard classes are defined in legends at the bottom of the figure)**

### 3.4 Overall Effect of Levee Integration on Flood Volume in Japan

Finally, we quantified the influence of levee integration on flood volume reduction, considering all the major rivers in Japan, We calculated total flood volume from gridded flood storage data for return periods (RPs) ranging from 5 to 1000 years. The analysis was restricted to simulation grids that contained levees.

The analysis shows that, across all return periods, levee integration leads to a noticeable reduction in total flood volume within levee grids (Fig. 10, shaded area). At lower return periods (e.g., RP 10–100 years), flood volume under natural conditions significantly exceeds the levee scenario, with a distinct shaded area capturing the avoided volume. As the return period increases, the total flood volume grows in both cases.



The reduction remains between 10% and 15% across the range of return periods. This consistent flood attenuation underscores the levee system's effectiveness in delaying and reducing floodwater accumulation within unprotected regions.

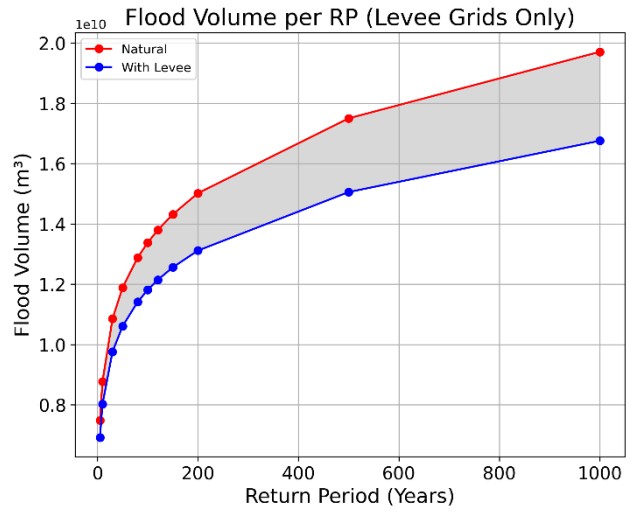

**Figure 10: flood volume trends and corresponding percentage reductions for each RP**

## 4    Conclusions

The volumetric downscaling approach applied in this study effectively captures the key characteristics of observed flooding. It enables the creation of high-resolution flood hazard maps from coarse-resolution flood volume data. The capacity to ascertain realistic flood behaviour at 1-arcsecond resolution, despite starting from 1-arcminute model outputs, underscores the critical role of downscaling in bridging the gap between large-scale hydrodynamic simulations and the high-resolution requirements of flood hazard mapping. This affirms that volumetric-based downscaling and levee schematisation provide a robust

framework for generating physically consistent, high-resolution flood hazard maps across complex river systems.

The integration of levee parameters through levee height and levee fraction into the flood modelling enhances the accuracy of flood extent estimation in both protected and unprotected areas. Nationwide simulations show that levee integration leads to a consistent 10–15% reduction in total flood volume across most return periods in levee-affected grids. At the unit catchment scale, reductions in inundated areas are even higher, depending on levee configuration and local topography. This implies that

the model with a volumetric downscaling framework can effectively represent the spatial variability of flood protection and convey both attenuation and overtopping dynamics.

The flood hazard maps generated for the Chikuma River basin reveal a clear progression in both inundation extent and severity with increasing return periods. While lower return periods (e.g., RP 10) already pose significant threats to urban areas and critical infrastructure, especially downstream of major confluences, the hazard footprint expands markedly by RP 100. Beyond



RP 100, the growth in flood extent becomes nonlinear. These patterns highlight the need for improved flood preparedness and infrastructure adaptation even under moderate flood scenarios and further stress the value of high-resolution, levee-aware mapping in informing such efforts.

Overall, the framework offers a practical and scalable approach to enhancing flood hazard mapping through the use of large-scale models. Future work may concentrate on incorporating dynamic levee failure and urban drainage features or components.
The proposed framework supports integrated flood management and climate adaptation planning in levee-protected and vulnerable floodplain areas.

## 5    Limitations

While the proposed framework demonstrates significant improvements in flood hazard representation, several limitations remain. First, the levee dataset, although derived from high-resolution LiDAR DEMs, may overlook smaller flood control
structures, leading to uncertainty in levee data and is subject to uncertainty in levee fraction estimation. Second, the frequency analysis used to estimate return-period flood volumes relies on limited sample lengths and may introduce bias at higher return periods. Third, the framework does not dynamically cater for levee overtopping or breaching; levee influence is treated as a static height barrier. Fourth, the input runoff data used in long-term simulations are daily averages, which may fail to capture short-duration peak flows, potentially underestimating flood magnitudes for certain return periods. Lastly, the current
framework only addresses fluvial flooding from river overflow and does not account for pluvial flooding due to intense rainfall and limited urban drainage capacity. These limitations suggest that future work should incorporate breach modelling and high-resolution urban drainage integration.

## CODE AVAILABILITY

CaMa-Flood (Yamazaki et al., 2011, https://doi.org/10.1029/2010WR009726) is open-source; the exact version (v4.2) used
for this study is archived at Zenodo (https://doi.org/10.5281/zenodo.11091435). All scripts required to reproduce our results, including the return period storage calculators for natural and leveed scenarios, the levee-aware downscaling routine, configuration templates, conda environment, launcher scripts, and figure scripts, etc., are archived at Zenodo (https://doi.org/10.5281/zenodo.17231818). The archive contains a step-by-step user instructions (REPRODUCE.md) to recreate the environment and regenerate all figures. For convenience, a code only (without trial data) is also available
(https://doi.org/10.5281/zenodo.17231022).

## DATA AVAILABILITY

Public and third-party datasets and access routes are as follows: GRADES hydrological forcing data (Yang et al., 2021, https://doi.org/10.1175/BAMS-D-20-0057.1), MERIT Hydro topography (Yamazaki et al., 2019, https://doi.org/10.1029/2019WR024873), the global levee dataset (Khanh et al., 2025,
https://doi.org/10.1029/2024GL114121). City flood hazard/hazard maps, Flood inundation extents from 2019 Typhoon



Hagibis and other data used for comparison and generating figures are provided in Zenodo archive (https://doi.org/10.5281/zenodo.17231818) , which includes detailed instructions (REPRODUCE.md) for environment setup, storage computation, downscaling, and figure reproduction. The Japan 1-arcsecond river network and floodplain maps used here are restricted; access can be requested from Dai Yamazaki (Institute of Industrial Science, The University of Tokyo) via

the project page https://hydro.iis.u-tokyo.ac.jp/~yamadai/cama-flood/. Sample outputs generated using these high-resolution maps (for figure reproduction) are included in the Zenodo archive (https://doi.org/10.5281/zenodo.17231818).

**COMPETING INTERESTS**

The authors declare that they have no conflict of interest.

**AUTHOR CONTRIBUTIONS**

MHA conducted the CaMa-Flood simulations, developed the storage-based downscaling code, performed all data analysis, and led the writing of the manuscript. YH administered the research, provided continuous guidance throughout the project, and contributed to the critical review and editing of the manuscript. DY provided the CaMa-Flood model and high-resolution river network data used in the simulations. GZ and YK developed the levee scheme in CaMa-Flood and offered initial guidance on levee integration. DNK gave the levee dataset used to generate the levee parameters incorporated into the model and helped

optimise the downscaling code.

**FINANCIAL SUPPORT**

This research was supported by the LaRC-Flood project co-funded by the New Energy and Industrial Technology Development Organization (NEDO) [Grant Number JP21500379], titled "Creation of Wide-Area Flood Risk Information Using Climate Model Output and Geographic Information Big Data". Additional support was provided by the Japan Science and Technology

Agency (JST) under the Moonshot R&D Program [Grant Number JPMJMS2281].

**ACKNOWLEDGEMENTS**

Observed river discharge data were obtained from the Ministry of Land, Infrastructure, Transport and Tourism (MLIT), Japan. Mapping and visualization were primarily performed using Python libraries, with supplementary spatial processing conducted in QGIS. Base maps include OpenStreetMap data © OpenStreetMap contributors. Some of the figures were produced using

the Scientific Colour Maps (Crameri, 2018). The authors also used ChatGPT (OpenAI) to improve the clarity and grammar of the manuscript text. The authors carefully reviewed and verified all AI-assisted edits to ensure accuracy and appropriateness.

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
