# Peer review of "Flood Volume Allocation Method for Flood Hazard Mapping Using River Model with Levee Scheme"

_EGUsphere, 2025_

## Referee Comment (RC2)

Review of Flood volume allocation method for flood hazard mapping using river model with levee scheme (egusphere-202504358).

Overprediction of flood inundation within levee protected floodplains by large-scale to global river models used in flood hazard and risk assessment is an important issue which needs to be corrected. The methodology here provides a useful framework of constraining flood hazard assessments along leveed river segments which employ global or large-scale river inundation models. However, the current manuscript would benefit from technical and editorial clarifications before being considered for publication. Please find my suggestions and recommendations below.

**General Comments:**

- 1. The manuscript would benefit from some discussion on which type of rivers and floodplains this framework would be most applicable to. Given the test area for this model is rivers in the country of Japan, the conceptual model which is implicit in this approach is the quantification of relatively confined floodplains (i.e., hundreds of meters to a few km wide). The authors have not demonstrated or explained if their modeling framework would be applicable to levee floodplains which are only semiconfined or unconfined (i.e. 10s to 100s km wide). Some discussion in the introduction or discussion sections of this manuscript on this issue would help the reader better understand the potential scale limitations of the proposed framework.
- 2. The filling and draining of floodplains commonly generate a hysteresis phenomenon where water levels and consequently flood inundation extents, depths, and presumably volumes for the same discharge are greater on the falling limb than the rising limb of the flood hydrograph. While the authors state they employed the annual maxima volumes in their study, it would be helpful to mention how they ensured "maxima volumes" were achieved.

**Specific Comments:**

- 1. Line 26 recommend changing high to large and then qualifying what the authors consider moderate (i.e., 10-year) to larger (>100-year) flood events
- 2. Line 59 GCM acronyms first use, recommend spelling it out so it is clear to the reader what GCM stands for (Global Climate Model [GCM]?).
- 3. Somewhere in section 2.3 it would be useful to mention why the Khanh et al., 2025 global levee data set was used over a potentially higher resolution national to regional levee dataset.

- 4. Somewhere near line 135 it may be worth noting that return periods for levee exceedances (over topping) do not equate to return periods for flood stages (levels) because levees are commonly constructed with freeboard as a factor of safety. In the U.S. this "freeboard" is commonly about a meter for levees with a designed exceedance of 1% annual chance flood (100-year flood). Consequently "100-year levees" will often not be overtopped by the flood stages/levels even with a < 0.2% annual chance exceedance probability.
- 5. Figure 3. Please check the x-axis units on the storage curve chart because m³ is too small for most floodplain storage volumes.
- 6. Figure 4. The figure description could use more robust description of the downscaling methodology shown here.
- 7. Line 231. Flood control has generally been replaced by flood risk reduction or mitigation. Philosophically, floods cannot be "controlled"; only the risk they pose to communities can be mitigated.
- 8. In line 105 the authors define the scenarios assessed a with and without levee. Then in line 239 they switch the without levee scenario to "natural scenario". Natural scenario, case, or simulation is used in lines 246, 247, 287, 297, 304, 332, 376 and the caption for Figure 8. I recommend sticking to without levee scenario, case, or simulation.
- 9. Line 288. I recommend using another word instead of significantly reduced as there was no statistical test performed to assess the significance of the reported reduction.
- 10. Lines 298 to 300. To my understanding, the modeling framework presented in this study does not evaluate structural performance of the levee which would require a geotechnical assessment of the structure. The ability of the levee to limit inundation for large magnitude floods (500- to 1000 Year RI) is a function of how the levee is parameterized in this modeling framework (over topping or no overtopping). If geotechnical levee dynamics and/ or a probabilistic breaching model were applied to the levee along the Chikuma River, then a statement of "structural performance" could be made.
- 11. It is also unclear what point the authors are trying to make in lines 300 to 301.
- 12. Lines 303 to 306. If you are going to make appoint about a spatial variability of levee "effectiveness" you need to show the figure supporting this statement in the main paper.
- 13. Line 335. I recommend clarifying that the authors are referring to a reduction in flood volume in this sentence.
- 14. Line 335-337. It is unclear what point the authors are trying to make in this sentence. This consistent flood attenuation underscores the levee system's

effectiveness in delaying and reducing floodwater accumulation within unprotected regions. There is a substantial amount of research showing levees increase flood levels and the extent of inundation across up stream or neighboring unprotected areas of the floodplain relative to a levee condition. Please clarify the point the authors are trying to make in this sentence.